# Calcitonin Gene-Related Peptide-Mediated Trigeminal Ganglionitis: The Biomolecular Link between Temporomandibular Disorders and Chronic Headaches

**DOI:** 10.3390/ijms241512200

**Published:** 2023-07-30

**Authors:** Linda Sangalli, Bradley Eli, Sachi Mehrotra, Suzan Sabagh, James Fricton

**Affiliations:** 1College of Dental Medicine—Illinois, Midwestern University, Downers Grove, IL 60515, USA; 2Facial Pain Specialists, San Diego, CA 92121,USA; drbradleyeli@gmail.com (B.E.); drsachi@facepaindocs.com (S.M.); drsuzan@facepaindocs.com (S.S.); 3Division of TMD and Orofacial Pain, University of Minnesota Schoof of Dentistry, Minneapolis, MN 55455, USA; 4Minnesota Head and Neck Pain Clinic, Plymouth, MN 55447, USA

**Keywords:** temporomandibular disorders, headache, migraine, central sensitization, calcitonin gene-related peptide

## Abstract

A bidirectional causal relationship has been established between temporomandibular disorders (TMDs) and chronic headaches. Recent advances in the neurobiology of chronic pain offer a framework for understanding the comorbidity between these two conditions that might reside in the shared biomolecular mechanisms of peripheral and central sensitization. The initiation of these processes is inflammatory in nature and is most likely mediated by key molecules, including calcitonin gene-related peptide (CGRP). This scoping review proposes that CGRP-mediated neuroinflammation in the trigeminal ganglion may partly explain the biomolecular bidirectional link between TMDs and chronic headaches. Finally, clinical implications of this neuropathologic process are briefly discussed.

## 1. Introduction

Temporomandibular disorders (TMD) encompass a group of conditions involving the masticatory muscles, the temporomandibular joint (TMJ) and associated structures [1,2]. TMD affect 5–12% of the population, with an estimated annual cost of approximately USD 4 billion in the US [2]. Approximately 15% of these individuals experience delayed recovery and chronicization of their condition [3].

Among the various signs and symptoms that individuals with TMD manifest, headache is a common manifestation. Migraine and tension-type headache constitute the majority of the primary headaches encountered in the general population, with a prevalence of approximately 12% [4] and 45–78% [5], respectively. Moreover, a new disorder has been introduced in the Diagnostic Criteria for Temporomandibular Disorders (DC-TMD), headaches attributed to TMD (HA-TMD), defined as “a headache in the temple area secondary to pain-related TMD that is affected by jaw movement, function, or parafunction” [6]. According to the diagnostic criteria, HA-TMD should be replicable with provocation and palpation of the masticatory system [5].

Recent studies highlighted the comorbid presence of headache and TMD, hypothesizing that a relationship between the two might exist. In this regard, a recent systematic review and meta-analysis supported a positive association between pain-related TMDs, migraine and chronic tension-type headaches [7]. Other studies observed that, among those patients seeking care for headaches, the prevalence of those presenting with TMD symptoms ranged from 52 to 55% [8]. When segregated into painful vs. functional TMD, this prevalence increased to more than 80% [9]. Headache severity has also been associated with the number of TMD symptoms such that the prevalence of headache in TMD patients with one TMD symptom was 57%, with two symptoms was 65%, and with three or more symptoms was 73% [10]. Moreover, longitudinal studies suggest that an increase in headache frequency may coincide with the development of TMD [11].

Importantly, the Orofacial Pain: Prospective Evaluation and Risk Assessment (OPPERA) study demonstrated that prior headaches were a significant risk factor for the development of first-onset TMD symptoms [8]. After the development of TMD, both headache severity and frequency were shown to increase in the TMD group compared to healthy participants. Specifically, the prevalence of migraine episodes increased 10-fold in the group that developed TMD symptoms [8]. Finally, clinical studies revealed that, when TMD is successfully treated, the headache symptoms also tend to resolve [12], suggesting a role of TMD in the pathophysiological process. Nevertheless, the biomolecular process that links these two conditions is far from being clear.

Recent advances in the neurobiology of TMD and headaches focused on the apparent bidirectional relationship and epidemiological association between the two conditions. At the center of these advances is an increased understanding of calcitonin gene-related peptide (CGRP), a mediator of trigeminal neuroinflammation [13], and its role in the process of pain chronicization. More specifically, CGRP is among the molecules linked to mediate a sterile trigeminal ganglionitis, wherein cross-excitation between different branches of the trigeminal nerve occurs [14]. When this ganglionitis is sustained, it leads to neuroplastic changes in the sensory cortex so that pain is perceived in the absence of noxious stimuli, a process referred to as *central sensitization* [15]. In this review, we will explore the biomolecular pathophysiology of headache (specifically migraine, tension-type headache and traumatic headache) and TMD as it relates to CGRP-mediated central sensitization, and briefly discuss how this understanding may guide treatment considerations.

## 2. Methods

A literature search was conducted to identify relevant studies examining the association between CGRP, headache (migraine, tension-type headache and post-traumatic headache) and TMD. The search aimed at including both preclinical and clinical studies investigating the role of CGRP in the pathophysiology of these conditions.

### 2.1. Search Strategy

Electronic databases, including PubMed, Embase and PsycINFO, were searched to identify relevant articles. The following keywords and terms were used: “Temporomandibular disorders”, “TMDs”, “TMD”, “migraine”, “tension-type headache”, “calcitonin gene-related peptide”, “CGRP”, “neuroinflammation”, “peripheral sensitization”, and “central sensitization”.

### 2.2. Study Selection Criteria

Included studies were studies (A) published in English; (B) both conducted on animal models and humans; (C) examining the relationship between CGRP, headaches and TMD; and (D) reporting relevant outcomes on CGRP, headache severity and TMD symptoms. Those not relevant to the topic, not published in English language and on animal studies without translational relevance were excluded.

The following sections report the main findings of this literature search.

## 3. CGRP in Peripheral and Central Sensitization

Both peripheral and central sensitization are cited as primary mechanisms behind the development of chronic pain syndromes, including headaches and TMDs. Peripheral and central sensitization encompasses two different phases, that are closely interconnected.

Peripheral sensitization. Peripheral injury of the muscles, joints, or nerves of the jaw and head triggers pain signals in the trigeminal nerve from the primary afferent fibers (mostly C fiber and A delta fibers). Local tissue inflammation releases cytokines and pro-inflammatory mediators, including CGRP, that perpetuate and increase the pain response. Peripheral sensitization lowers the depolarization threshold, so that normal stimulation is now perceived as painful (primary allodynia) and painful stimuli result in higher pain perception (primary hyperalgesia or “hyperalgesic priming”) [16,17].Central sensitization. Sustained peripheral pain signaling leads to central sensitization, characterized by increased excitability of central pain pathways [18] https://www.iasp-pain.org/resources/terminology/ (Accessed on 25 February 2023) [18,19]. At first, this sensitization is activity-dependent and consists primarily of lowered depolarization thresholds. This characterizes the phase of acute pain. However, if it persists for a longer period of time (i.e., beyond the normal healing process), it evolves into an activity-independent phenomenon through neuroplastic adaptation [14]. In this scenario, the CGRP released in the trigeminal ganglion engages with adjacent neurons and satellite glial cells, causing the continuation of peripheral sensitization and facilitating central sensitization of the second-order neurons [13]. This identifies a shift to a chronic pain phase. Central sensitization is the physiological hallmark of chronic pain syndromes and is responsible for the clinical symptoms of secondary hyperalgesia (defined as the increased pain response derived from a normally painful stimulus (Terminology | International Association for the Study of Pain. International Association for the Study of Pain (IASP). https://www.iasp-pain.org/resources/terminology/ (Accessed on 25 February 2023) [18]) and secondary allodynia (defined as pain response derived from a stimulus that is not normally perceived as painful [19].

As noted earlier, one pivotal molecule responsible for both states is CGRP. CGRP receptors belong to the G-protein coupled receptor (GPCR) family and are composed of two subunits, the calcitonin receptor-like receptor (CLR) and the receptor activity-modifying protein (RAMP) [20]. The CLR serves as the primary binding site for CGRP, while RAMP modifies the pharmacological properties of the receptor [20]. The expression of CGRP receptors varies among different types of sensory neurons within the ganglion. These receptors are predominantly expressed by peptidergic nociceptive neurons, which are responsible for transmitting pain signals [21]. However, they are also found in non-peptidergic neurons and certain subsets of proprioceptive and mechanoreceptive neurons [21]. The differential expression of CGRP receptors by various sensory neuron types contributes to their distinct responsiveness to CGRP signaling. This variation in receptor expression gives rise to both autocrine and paracrine signaling by CGRP within the ganglion [22]. Specifically, autocrine signaling occurs when CGRP acts on the same neuron responsible for its production [22]. In this scenario, CGRP released from the neuron can bind to the CGRP receptors present on its own cell membrane, thereby influencing its own cellular activities. Conversely, paracrine signaling involves the diffusion of CGRP to neighboring sensory neurons within the ganglion [22]. Accordingly, when CGRP is released from one neuron, it can travel short distances to interact with CGRP receptors expressed on nearby neurons [22]. This paracrine signaling allows for intercellular communication, affecting the excitability and sensitivity of adjacent neurons in the ganglion.

CGRP is abundantly distributed in the central and peripheral nervous system and pain pathways, as it is found in unmyelinated Aδ and C sensory nerve fibers [23]. Even if the attention brought to this molecule thanks to its therapeutical effect in migraine refer to the CGRP released in the brain, the primary source of CGRP is not within the brain itself, but rather in peripheral structures such as nerve endings and sensory ganglia. CGRP is predominantly release from peripheral nerve fibers, including those located in the trigeminal ganglion. Here, it is synthesized in the cell bodies of these sensory neurons [24] and then transported to their peripheral terminals. Nevertheless, it should be pointed out that this study applied a stimulus constituted by capsaicin directly into slices of the ganglion in vitro. As such, the notion that excitation of afferent fibers causes the release of CGRP from neuronal soma in the trigeminal ganglion may be speculative. Within the central nervous system (CNS), CGRP is also found in some regions where it likely acts as a neurotransmitter or neuromodulator. These regions include:Spinal cord, where CGRP is released from primary sensory neurons in the dorsal horn of the spinal cord and cerebral gray matter, where it contributes to pain transmission and modulation [21];Brainstem: CGRP-containing fibers and terminals have been identified in various brainstem nuclei involved in pain processing, including the periaqueductal gray (PAG) and the nucleus tractus solitarius (NTS);Hypothalamus: CGRP has been detected in certain hypothalamic nuclei, such as the paraventricular nucleus (PVN) and the supraoptic nucleus (SON), involved in the regulation of autonomic functions and pain modulation [21];Thalamus: neurons expressing CGRP in the parvocellular sub-parafascicular nucleus have been observed in the thalamus [25].It is also broadly distributed in non-neuronal tissues, such as mesenteric plexus, gastrointestinal, cardiovascular and nociceptive systems, smooth and skeletal muscles, and skin [21,26].

Although other possible mechanisms are still debated [27], CGRP is implicated as the primary activating factor of migraine headaches and TMD via cross-excitation with resulting stimulation and perpetuation of peripheral and central sensitization, both in the acute *activity-dependent* phase and in the chronic *activity-independent* phase. The concept of cross-excitation between sensory neuron cell bodies occurs among adjacent and long-distance neurons. To support this, early electrophysiological studies showed that stimulating one cell led to a partial depolarization of neighboring neurons in the dorsal root ganglion (DRG), thereby lowering the activation threshold of secondary neurons [28,29]. The phenomenon was hypothesized to occur through chemical communication [28,29]. Recent studies have further revealed that the cross-excitation of adjacent neurons in the DRG involves the participation of gap junctions and the propagation of Ca^2+^ waves through neurons and satellite glial cells [30]. This neuronal coupling significantly increases in models of inflammation or neuron injury, and contributes to the activation of certain cells by capsaicin. Although CGRP does not appear to play a mechanistic role in this specific mode of cross-excitation, it is possible (albeit speculative) that peptidergic neurons are involved or undergo cross-excitation. The fact that CGRP can diffuse over long distances suggests its potential involvement in the cross-sensitization of spatially distant neurons. This may differentiate neuropeptide-mediated coupling from the gap junction-mediated coupling observed in adjacent neurons.

Specifically in regard to migraine and TMD pathogenesis, review studies have suggested that pain-related TMD symptoms might be attributed to a peripheral mechanism in certain cases [31]. However, given that the correlation between the severity of TMD-related pain symptomatology and the evidence of tissue pathology is often relatively weak, some other patients might experience a central sensitization phenomenon. As a result, an alteration in the central nervous system pain processing pathways along with responsible heritable genes encoding for altered pain processing might be responsible for pain symptoms [31]. Among other important factors, biopsychosocial stressors are also known to play a role in development and chronicization of the painful condition [31].

With migraine pathogenesis, the role of CGRP in developing central sensitization has been established. Migraine pathogenesis is complex and can be summarized as a primary brain disturbance that involves ion channels, thus creating a neurovascular and neurobiological disorder where neural events result in dilation of blood vessels with subsequent pain and further nerve activation (10.4103/0972-2327.99993, #83). With tension type headache, the evidence suggests that an increased excitability of the central nervous system secondary to sustained and repetitive pericranial myofascial input and central sensitization may be implicated in the transformation of tension-type headache from episodic to chronic [32,33]. Conversely, the pathophysiology of traumatic headache is not well understood. Nevertheless, it seems to involve neurometabolic changes and an impairment in descending modulation, as well as an activation of the trigeminal sensory system, including peripheral and central sensitization [34].

The role of CGRP in peripheral and central sensitization is thought to be mediated by its neuroinflammatory and vasodilation effects. Among other molecules such as glutamate and prostaglandin E2, CGRP promotes the further release of nitrous oxide (NO) from postsynaptic neurons. As a result, this further sensitizes neurons through stimulation of inflammatory mediators in the periphery, the trigeminal ganglion, in secondary connections in the trigeminal nucleus caudalis, and tertiary connections in the thalamus, limbic system and sensory cortex (Figure 1) [13,35]. Moreover, experimental evidence showed that intrathecal administration of CGRP stimulates central supporting cells (microglial cells and astrocytes) to release pro-inflammatory mediators that are known to induce and perpetuate central sensitization [15].

## 4. CGRP and Migraine

Different migraine pathogenetic mechanisms have been explored and supported in the literature. Beside the classic vascular pathophysiology of migraine, current research suggests that vasodilation is an epiphenomenon, and that peripheral and central sensitization in the trigeminal nociceptive system may explain the migraine pain [36]. This paradigm shift was accompanied by an increased understanding of the role of CGRP in migraine pathogenesis, where CGRP functions as a potent vasodilator, a sensory neurotransmitter and a regulator of gene expression. Especially for its property of mediator of vasodilation, it is believed to cause neurogenic inflammation, while co-localized with several other biomolecular markers of pain [37].

The role of CGRP in migraine pathogenesis is evidenced by many of the current migraine pharmacological therapies, which act by directly or indirectly inhibiting CGRP activation. For instance:Ergotamine derivatives and triptans, drugs approved for the acute treatment of migraine [38], mainly stimulate 5-HT_1B/1D_ receptors. As 5-HT_1B_ receptor is localized on smooth muscle cells of cerebral, meningeal and coronary arteries, and 5-HT_1D_ is mainly expressed in the trigeminal ganglion, these drugs result in a strong vasoconstriction of the cranial arteries [39]. They also indirectly act on decreasing the release of CGRP, thus reducing trigeminal activation and vasodilation [35].Ditans, 5-HT_1F_ receptor agonists (lasmiditan) are newly Food and Drug Administration (FDA)-approved drugs for the acute treatment of migraine [40]. 5- HT_1F_ receptors are located on terminals and cell bodies of the trigeminal ganglion neurons, acting at the peripheral nervous system and central nervous system. It can modulate CGRP from trigeminal ganglion neurons by potentially blocking its release and inhibiting the development of central sensitization [40]. Contrary to the effect of ergotamine derivates and triptans, activation of 5- HT_1F_ does not induce vasodilation but rather causes vasoconstriction [40].CGRP receptors are localized on smooth muscles cells of meningeal and cerebral blood vessels. As a result, the release of CGRP by the activated meningeal C-fibers causes blood vessels to dilate [37]. Direct blockade of CGRP signaling with gepants [41], and with monoclonal antibodies directed against the molecule or its receptor attenuated the cutaneous mechanical hypersensitivity [42] and nitroglycerin-induced trigeminal hyperalgesia in animal models of migraine-like pain [43]. They have been FDA-approved as an effective treatment in preventive migraine [44]. Although the exact mechanism and site of action of CGRP in pain are still unclear and many mechanisms of action have been proposed [45], the meningeal blood vessels and their vasodilation are a primary target to prevent or inhibit pain signals [23].Topiramate inhibits nitric oxide and proton mediated CGRP secretion in a time- and concentration-dependent fashion from sensory trigeminal neurons [46].Botulinum toxin-A (BoNT) at doses between 150 Units and 195 Units, repeated every 12 weeks, is listed among the FDA-approved therapies for prevention of chronic migraine [38]. Among the several hypotheses on mechanism of action, one of these supports that BoNT attenuates the release of local transmitters such as CGRP from trigeminal neurons [47]. This further supports the pivotal role of CGRP reduction to its mechanism of action [48].

Nevertheless, as many as 40% of patients suffering from chronic migraine are considered non-responders to therapy, i.e., not achieving more than 50% reduction in monthly headache days [49]. This may indicate that, while CGRP plays a pivotal role in the pathology for many cases, it may not be universally applicable, or not reflect the only molecule or mechanism of action involved in migraine pathogenesis. Emerging evidence suggests that combining BoNT with anti-CGRP antibodies or anti-CGRP receptor antibodies results in a slight improvement compared to monotherapy [50,51]. However, non-responder rates still remain high. Moreover, reports indicate that the success of BoNT treatment correlates with high GRP levels in patients’ serum, although with limited available data and with other studies supporting opposite findings [52,53]. These findings may suggest that CGRP assays could potentially aid in assessing patient suitability for anti-CGRP therapy. Nevertheless, the costly, time-consuming and technically demanding existing assay technology may limit its suitability.

Important experimental clinical findings supporting the relationship between migraine and CGRP include the following findings:The levels of CGRP in saliva, cerebrospinal fluid and peripheral blood are elevated during severe migraine attacks [14].Clinical improvement of migraine symptoms after triptan administration is accompanied by decreased CGRP levels [54]. In addition, administration of CGRP triggers migraine headaches in both healthy subjects and individuals suffering with migraine [55,56].Subjects with chronic migraine have higher levels of CGRP in peripheral blood samples than those with episodic migraine [57].

## 5. CGRP and TMDs

TMD exhibits a significant comorbidity with migraine. Current research supports that CGRP may constitute one important molecular link between headaches and TMD. Such a connection is evidenced by a migraine animal model, where the masseter muscle was injected with a chemical irritant to trigger a peripheral inflammatory response [58]. As a result, this inflammatory response led to a rapid and significant increase in CGRP mRNA in the trigeminal ganglion [58]. Thus, this increased CGRP expression provoked migraine-like behavioral phenotypes of hypersensitivity [59]. It is hypothesized herein that the findings of this experiment may be explained by cross-excitation within the trigeminal ganglion. Trigeminal sensory neurons are pseudo-unipolar neurons, which refer to the fact that trigeminal neurons have a single process emerging from the cell body that splits into two branches—a central branch towards the CNS, and a peripheral branch that travels towards the target tissue or sensory receptor. The peripheral afferents of both meningeal tissues and the TMJ apparatus project towards the trigeminal ganglion location of their cell bodies. Hence, the trigeminal ganglion is the common area where both meningeal and temporomandibular sensory neurons project. Secondary neurons from the trigeminal ganglion then project to the trigeminal nucleus caudalis, which in turn project to the thalamus, cortex and limbic system [60]. Once CGRP is released from neuronal cell bodies or processes, it further contributes to the inflammatory process by promoting sensitization of surrounding neuronal and glial cells not initially involved in the inflammatory response [14].

Specifically in the case of TMJ-related trauma, peripheral insult or repetitive strain to the masseter causes CGRP expression in the trigeminal ganglion. When CGRP is expressed in the ganglion, it acts in an autocrine and paracrine fashion by stimulating satellite microglial cells to release pro-inflammatory mediators, causing a ganglion-wide inflammatory cascade [61]. In this regard, one study showed that administering CGRP to microglial cells in a culture stimulated the release of over 20 inflammatory cytokines [62]. This neuroinflammatory cascade explains how peripheral inflammation of trigeminal afferents from the mandibular branch of the trigeminal nerve can stimulate migraine-like behavioral phenotypes [14]. This sequence is also applicable in the opposite direction, as evidenced by the fact that migraine is a significant predictor of TMD onset [8] or that cases of post-traumatic brain injury can lead to TMD-like symptomatology [63].

The role of CGRP as a key biomolecular link in neuroinflammation responsible for TMD development and maintenance is highlighted in the following studies (Table 1).

The majority of the available literature on the use of anti-CGRP therapy in TMD has been conducted in animal studies. Interestingly, there is evidence suggesting a potential association between comorbidity of headache, migraine and TMD, and dysfunctional CGRP signaling. Studies have indicated alterations in CGRP levels and CGRP receptor expression in individuals with both headache and TMD compared to those with either condition alone. Comorbidity of headache and TMD could potentially serve as an indicator for the likelihood of successful anti-CGRP therapy [59,70]. While specific evidence addressing this question is limited, ongoing randomized clinical trials are currently investigating the potential TMD pain reduction derived from administration of monoclonal antibodies against CGRP receptors (https://clinicaltrials.gov/ct2/show/NCT05162027 and https://clinicaltrials.gov/ct2/show/NCT04884763?term=CGRP&cond=Temporomandibular+Disorder&draw=2&rank=2) (Accessed on February 23 2025).

When these findings are combined with those supporting the role of CRGP as a biomolecular key in headaches, a framework to better understand the shared pathogenesis of headache and TMD comorbidity emerges. This relationship is supported by the following findings:CGRP receptors are widely distributed through the trigeminovascular system [71].The inflammatory response from peripheral injury stimulates the expression of CGRP in the trigeminal ganglion and central relay centers [13].Biomolecular CRGP release in the trigeminal ganglion stimulates the release of pro-inflammatory mediators via supporting cells in a paracrine fashion [72].The resulting “inflammatory soup” in the ganglion (a sterile ganglionitis) is permissive of cross-excitation of all branches of the trigeminal nerve [73].

Taken together, these findings suggest that cross-excitation secondary to CGRP-mediated trigeminal ganglionitis may explain the physiologic basis for the comorbidity of TMD and headaches [15].

## 6. CGRP, Traumatic Headaches and TMD

The proliferation of research on mild traumatic brain injury (mTBI) and post-traumatic headaches (PTH) in the past decade has increased our understanding of chronic pain syndromes. There are many types of trauma that can impact the head, neck and jaw, and lead to both headache and TMD including:Direct jaw trauma;Sports injuries;Motor vehicle accidents;Whiplash associated injuries;Hyperextension injuries;Strain from repetitive or continuous muscle activation;Bruxism and other parafunctional behaviors.

Importantly, an acute jaw, head and neck injury of any type will result in a four-fold increase in the odds of subsequent painful TMD development [8]. Trauma to the TMJ complex is frequently overlooked in the acute setting or, if noted, the prescribed treatment is usually limited to a soft diet recommendation. This contrasts with other orthopedic injuries, where a program of stabilization, physical therapy and other supportive care is reflexively prescribed.

There are well described clinical and phenotypic overlaps between PTHs and both migraine [74] and TMDs [63]. This is because persistent post-concussive symptoms like PTH follow the same process of central sensitization as TMD, migraine headaches and a host of other chronic pain conditions [75].

Not surprisingly, CGRP is implicated in the development of PTH after mTBI [76]. According to preclinical studies, using CGRP inhibitors in the first week after mTBI nearly eliminated the development of post-traumatic hyperalgesia in preclinical models [77]. Nevertheless, this effect was reduced when CGRP inhibitors were administered 1–2 weeks after the injury, indicating that the timing of CGRP blockade is essential to prevent the development of central sensitization [78]. Similar findings were observed with the administration of BoNT therapy, which was able to prevent the development of PTH when given within the first week after the injury. Similarly, this positive effect exhibited limited effects when given after the acute period [79].

Another recent study supporting the link between CGRP and PTH revealed how the injection of CGRP in a cohort of patients diagnosed with PTH resulted in the immediate development of migraine-like headache in 72% of them [80]. This study further provided a rationale for the use of anti-CGRP monoclonal antibodies as a treatment for PTH. Similar clinical trials are ongoing, suggesting positive initial results [80].

## 7. Treatment Considerations

The studies reviewed so far suggest that an anti-neuroinflammatory treatment in the acute phase of an injury has the potential to encourage healing and prevent central sensitization, and, as a result, a chronic pain state. Alternatively, if a chronic pain condition is already established, the same treatment strategy can be used in an abortive fashion for episodic symptom flare-ups. In this scenario, pharmacologic agents that directly or indirectly antagonize CGRP are already being used clinically for different types of headache [81,82]. Current research also supports the use of CGRP antagonists for TMD-related pain [69]. Nevertheless, at the current state of art, an application of anti-CGRP therapies to be provided as an early therapeutic strategy may not be feasible. First of all, a reason why these therapies may not be initiated as early interventions include current guidelines and limited approval. To date, anti-CGRP therapies are primarily approved for the preventive treatment of migraine, and their use as early interventions for acute migraine attacks or TMD is still an area of ongoing research and clinical evaluation. Moreover, treatment guidelines and recommendations may not yet include anti-CGRP therapies as front-line or even second-line treatments, leading to their prescription primarily for patients who have failed initial treatment options. For example, anti-CGRP monoclonal antibodies are approved by health insurance in the United States after documented inability to tolerate or failure of an 8-week trial of two or more other Level A or B migraine treatments [38]. In addition, while these therapies have consistently demonstrated efficacy in reducing the frequency and severity of migraine attacks in clinical trials, their effectiveness in an acute setting or the early stages of migraine and TMD is still being studied. As such, specific mechanisms of action, dosage and optimal timing for these therapies need further investigation to establish their potential as early interventions. Similar caution should be taken in regard to safety and side effects. To date, anti-CGRP therapies have shown favorable safety profiles in clinical trials while being used as a preventive treatment. Common side effects reported include injection site reactions, constipation and upper respiratory tract infections [38]. However, comprehensive data on long-term safety and potential off-target effects may still be evolving. Finally, these drug therapies are relatively expensive compared to other treatment options for migraine and TMD, which may limit their accessibility and making them less feasible as early interventions.

Therefore, the optimal approach in clinical practice should involve an early identification of those individuals in the acute phase with a high likelihood of developing a chronic pain state through the promotion of risk stratification and preventive treatment approaches. At this stage, conservative, non-invasive and inexpensive approaches are advocated to encourage both healing of the injured tissues and reduction of the risk factors that drive delayed healing and chronic pain.

The first few weeks after an injury correspond well with the proposed physiological mechanisms that drive central sensitization, i.e. neuroinflammation, microglial activation and neuroplastic adaptation. This is also supported by clinical findings, such as those seen in the OPPERA study [2]. Each individual case is different, but the principle of *prompt treatment* to prevent central sensitization is clear. If central neuroplastic changes can be prevented, chronic pain will not develop. A multimodal approach, using non-invasive and complementary measures, provides the best chance of curbing the cycle of pain amplification and progression to pain chronicity. 

Table 2 lists the characteristics and scientific efficacy of different treatments of TMD and headache conditions [83,84,85,86,87].

Beside pharmacological approaches, other non-pharmacological strategies can be adopted. This is the case, for example, of an anti-inflammatory minimal sugar diet with nutritional supplements that attenuate biomolecular CGRP expression [88,89] or the use of nutritional supplement in patients suffering from migraine (such as coenzyme Q10 alone [90,91,92] or in combination with nano-curcumin [93], L-carnitine [94], curcumin [95] and vitamin D [96], among others). Notably, daily supplementation with coenzyme Q10 [92], curcumin [95] and vitamin D [96] were shown to reduce CGRP levels. Some other evidence derived from preclinical migraine and TMD model studies reported reduction of CGRP secretion with grape seed extract [97,98,99]. Table 3 summarizes the evidence of the efficacy and scientific rigor of the available human studies assessing nutritional supplements compared to a control group.

In cases of pain within the temporomandibular structures, the protective biomechanical splinting of the TMJ and jaw and head muscle complex is a non-invasive effective intervention. As noted, there is a large population of CGRP-positive neurons in the TMJ complex (joint capsule, synovial membrane, retrodiscal tissue), and masticatory muscles, tendons and fascia [69,102,103]. Tooth clenching can activate these neurons through repetitive strain of masticatory muscles and TMJ tissues, especially if the disc space was recently traumatized. Sustained clenching even for short periods can trigger pain, and lead to peripheral sensitization in the muscles of mastication [104] with nociceptive sensitization and tissue oxygenation changes [105]. Oral appliances, such as an anterior bite plane which impedes posterior occlusal contact, can mechanically reduce noxious trigeminal input through reduction of clenching forces, and overload of TMJ and masticatory muscles, especially during the acute stage [74,106].

The use of behavioral interventions (such as mindfulness strategies [79] and deep breathing techniques [83,107,108] to reduce peripheral and central sensitization may also play a role in managing acute and chronic pain symptoms by decreasing psychosocial stressor and harmful oral parafunctional habits. Interestingly, expression of CGRP can induce anxiety-like behaviors in clinical models [109]. Further, experimentally induced masseter myalgia has been shown to activate CGRP release in the hypothalamic–pituitary–adrenal axis, establishing a direct link with stress levels [110]. As CGRP-positive neurons are also abundantly present in the “nociceptive amygdala”, the brain region that integrates affective and sensory signals [111], targeting psychological distress may represent an indirect pathway to intervene in CGRP-mediated pain chronicization.

Physical therapies and exercise are considered front-line therapy for acute head, neck or TMD pain. Early mobility of both the jaw and neck is important for the interruption of peripheral sensitization. Multiple studies have shown that jaw exercises facilitate pain-free mouth opening and reductions in disturbances to daily living [19]. Physical therapies such cryotherapy (cold therapy) or thermotherapy are a well-established treatment for acute orthopedic injuries. Cold decreases circulation to the injured region, slowing the inflammatory reaction and reducing inflammation. Cold has an analgesic effect by slowing nociceptive conductivity, decreasing neuronal excitability and decreasing muscle tension. Thermotherapy (heat therapy) is also well-established and is typically used in a delayed fashion several days after injury (to avoid bleeding into the wound). Thermotherapy increases circulation and oxygenation, and allows for the elimination of metabolic waste. Heat also reduces nerve pain conduction. Finally, heat promotes muscle relaxation, facilitating ease of mobility and the reduction of the guarding reflex.

## 8. Conclusions

Headache and TMD pathology are linked via the trigeminal system. The frequent comorbidity and bidirectional relationship of these two conditions may be supported by biomolecular pathways such as CGRP-mediated cross-excitation in an inflamed trigeminal ganglion. As a result, this excitation may drive peripheral and central sensitization of the structures innervated by the trigeminal system including both TMD and headache syndromes such as migraine. In both the acute and chronic phase of these conditions, treatments that inhibit CGRP may represent a promising preventive treatment strategy to minimize delayed recovery and chronic pain. This also highlights the importance of multimodal treatment with conservative, non-pharmacologic measures during the acute period including dietary changes, oral device therapy, behavioral management and physical therapies such as exercise to reduce peripheral and central risk factors to prevent chronic pain.

## Figures and Tables

**Figure 1 ijms-24-12200-f001:**
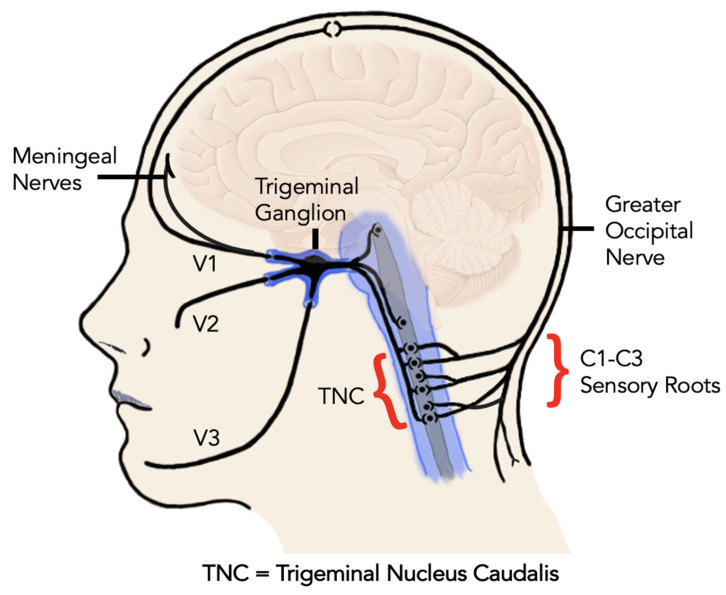
The neuroanatomy of the trigeminocervical complex including the ophthalmic nerve (V1) innervating the eyes, upper eyelids and forehead, the maxillary nerve (V2) innervating the cheeks, nose, lower eyelids, upper teeth, lips and gingiva, and mandibular nerve (V3) innervating the lower face, jaws, lower lip, lower teeth and gingiva.

**Table 1 ijms-24-12200-t001:** Studies supporting a biomolecular link between CGRP and TMD in animals and humans.

Preclinical Studies		
Author	Model	Findings
Cady et al. [15]	TMJ capsule	Injection of CGRP in the TMJ capsule stimulated the expression of proteins associated with peripheral and central sensitization in neuronal and glial cells in animal models.
Fiorentino et al. [64]	Osteoarthritis in a mouse model	CGRP-induced neuroinflammation contributed to histopathological modifications of the articular tissues (i.e., cartilage), leading to osteoarthritis in a mouse model.
Lai et al. [65]	TMJ of mouse model	Inhibition of CGRP-mediated neuroinflammation curbed the progression of TMJ damage.
Akerman et al. [59]	Rat model of myofascial TMD-like inflammation	Models of TMD-like inflammation resulted in neuronal activation and sensitization of dural trigeminal neurons, similar to migraine-like manifestation. Pre-administration of CGRP receptor antagonist effectively prevented these neuronal responses.
Brouxhon et al. [66]	TMJ of mouse model	The overexpression of CGRP in mouse models of TMJ led to the manifestation of joint anomalies and articular pathology; conversely, in a scenario of joint inflammation, the overexpression of CGRP inhibitory peptide partially led to improvement of joint pathology.
Shu et al. [67]	Myogenic TMD mice model	The presence of pre-existing myogenic TMD lesions caused increased central CGRP release and enhanced migraine hypersensitivity in animal models.
Damico et al. [68]	Acute and chronic arthritis model	Both acute and chronic arthritis were associated with significant increases in CRGP expression in the trigeminal ganglion in animal models.
Suttle et al. [69]	Mouse model	In naïve mouse models, the local injection of CGRP in masseters and/or TMD induced acute pain. Conversely, blockage of CGRP receptor decreased TMD pain.
Romero-Reyes et al. [70]	Mouse model of acute masseter pain	Selective CGRP receptor antagonist, MK-8825, was found to significantly reduce spontaneous orofacial pain behaviors in a mouse model of acute masseter pain injected with CFA. This study also supported the role of CGRP as an important neurotransmitter involved in TMD pain, although not through an inflammatory mechanism.
Clinical studies		
Sato et al. [44]	TMJ pain vs. healthy control	Human subjects exhibited a significantly higher level of CGRP in deranged TMJ joints vs. healthy controls, with CGRP levels that are positively correlated with pain intensity scores.

**Table 2 ijms-24-12200-t002:** Characteristics and scientific basis of treatments for TMD and headache.

Intervention	Scientific Basis	Description
Self-management training	Systematic reviews of behavioral therapies	Nutritional and dietary intervention Preventive medicine counseling Habit-reversal Mindfulness-based stress reduction Meditation and relaxation
Intra-oral splints	Systematic reviews of intra-oral splints	Full coverage stabilization at night Repositioning splints at night Immediate quick splints short-term Anterior bite plane short-term
Medications	Systematic reviews of medications	Migraine medication NSAIDs Acetaminophen Tricyclic medications Muscle relaxants
Physical therapies	Systematic review evidence of therapeutic exercises	Therapeutic exercises Mobilization

**Table 3 ijms-24-12200-t003:** Characteristics and scientific evidence of nutritional supplement in patients with migraine.

Nutritional Supplement	Scientific Rigor	N Participants	Findings
Coenzyme Q10	Double-blind placebo-controlled RCT [92]	45 female adults with migraine	Significant reduction in frequency (*p* = 0.018), headache intensity (*p* = 0.001) and duration (*p* = 0.012) compared to controls
	Open-label match-controlled trial [90]	80 adults with migraine	Significant reduction in frequency of monthly attacks (*p* < 0.001) and headache severity (*p* < 0.001) compared to controls
	Crossover double-blind placebo-controlled RCT [91]	120 children and adolescents with migraine	Greater improvement in migraine frequency in the initial 1–4 weeks
	Double-blind RCT [100]	42 adults with migraine	Significant decrease in attack frequency, headache-days and responder rate (47.6% vs. 14.4% in controls)
Nano-curcumin and coenzyme Q10	Double-blind placebo-controlled RCT [93]	100 adults with migraine	Significant reduction in headache frequency, severity and duration in participants treated with nano-curcumin and coenzyme Q10 (*p* < 0.001)
L-carnitine and coenzyme Q10	Double-blind placebo-controlled RCT [94]	56 adults with migraine	Significant reduction in headache intensity (*p* < 0.001), duration (*p* < 0.001), frequency (*p* < 0.001) and headache diary results (*p* < 0.001)
L-carnitine, magnesium and magnesium-L-carnitine	Single-blind RCT	133 adults with migraine	Magnesium supplementation achieved significantly higher reduction in headache frequency compared to the other groups (*p* = 0.008); significant reduction in migraine symptoms in all study groups with no difference among them
Curcumin	Double-blind placebo-controlled RCT [95]	44 female adults with migraine	Significant reduction in headache intensity (*p* = 0.001) and duration (*p* = 0.007); no significant reduction in headache frequency (*p* = 0.052)
Vitamin D	Double-blind placebo-controlled RCT [96,101]	80 adults with migraine	Significant reduction in migraine disability (*p* = 0.016), headache duration, intensity and frequency (*p* < 0.05) compared to controls

RCT: randomized clinical trial.

## Data Availability

Not applicable.

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
