# Peer review of "Calcitonin Gene-Related Peptide-Mediated Trigeminal Ganglionitis: The Biomolecular Link between Temporomandibular Disorders and Chronic Headaches"

_ijms, 2023, doi:10.3390/ijms241512200_

Round 1

Reviewer 1 Report

The report is in the attachment.

The quality of English is very good, with only minor changes needed.

Author Response

Response to Reviewer 1

Reviewer: This review explored commonalities between the pain mechanisms involved in temporomandibular disorders (TMD) and chronic headaches. A high level of comorbidity for these conditions has been noted, and in this review reasons for this are proposed. It is suggested that there is a bi-directional role in comorbidity for the cross-excitation of neurons of the trigeminal ganglia due to their somatic release of CGRP, which contributes to both peripheral and central sensitisation of the trigeminal system. This is an interesting new twist on an idea suggested in one of the cited references and it would be of interest to the IJMS readership.

The paper gives a comprehensive overview of the emerging evidence of a central role for CGRP in migraine, arising from the success of pharmacological treatments that antagonise CGRP signalling by targeting the neuropeptide, its receptor or signalling pathways linked to CGRP production. It then proceeds to discuss the evidence that CGRP is involved in TMD, which has encouraged the establishment of two ongoing clinical trials examining the potential for reduction of TMD pain by monoclonal antibody therapies against the CGRP receptor. These initial sections are very well written with easy to follow and convincing arguments. However, there are some topics raised in the review that merit a more in-depth discussion whilst there are other sections that are not well integrated into the overall theme of the article.

Answer to the reviewer: We thank the reviewer for the thorough assessment of our paper and for the encouraging and positive words of appreciation. We now provided a revised version according to the proposed suggestions, that we hope may address the reviewer’s concern.

Reviewer: Specific comments

There is an over-reliance on the citation of other reviews where often it would be better to refer to the primary studies; for example, when citing specific effects of CGRP (Ln 94 to 121, 212-221) or botulinum toxin A. Regarding the latter, there are numerous published works that report its blockade of CGRP release from trigeminal neurons that seem more appropriate than the cited reference [21]. In view of a central hypothesis of the review being that CGRP release from neuronal cell bodies causes cross- sensitisation of other neurons in the ganglia, based on a proposal in a cited review [13], the original research showing that neuronal soma of trigeminal ganglia release CGRP should also be cited (Ulrich- Lai et al., 2001, Pain 91, 219-226). It should be pointed out that this study applied a stimulus (capsaicin) directly onto slices of the ganglion in vitro, so the notion that excitation of afferent fibres causes the release of the neuropeptide from neuronal somata in the ganglion is speculative.

Answer to the reviewer: Original experimental studies have been added, such as (A) Christensen et al, Cephalalgia 2019 to support the role of CGRP monoclonal antibodies to reduce cutaneous hyperalgesia in animal models with migraine-like pain. (B) Christensen et al, Cephalalgia 2020 to support the notion that a clear mechanism of mechanism and site of action of CGRP monoclonal antibodies and gepants is still debated. (C) Greco et al, Cells 2022 to support the statement that CGRP monoclonal antibodies attenuated nitroglycerine-induced trigeminal hyperalgesia in formalin test in animal model of chronic migraine. (D) Edvinsson et al, J Headache Pain 2022 to support the expression of 5-HT1B/1D as target of triptans. (E) Hansen et al, Cephalalgia 2010 to provide evidence that administration of CGRP triggers migraine attacks. (F) Ho et al, Neurology 2014 to support the role potential role of gepants as CGRP receptor antagonist in migraine prophylaxis. (G) Durham et al, Headache 2006 to support that release of CGRP is attenuate by administration of topiramate.

In addition, the original paper showing that neural soma of trigeminal ganglion release CGRP has been quoted (page 3, section 3. CGRP in Peripheral and Central Sensitization), and now reads ;“Here, it is synthesized in the cell bodies of these sensory neurons (Ulrich-Lai,  #94) and then transported to their peripheral terminals. Nevertheless, it should be pointed out that this study applied a stimulus constituted by capsaicin directly into slices of the ganglion in vitro. As such, the notion that excitation of afferent fibers causes the release of CGRP from neuronal soma in the trigeminal ganglion may be speculative.”

Lastly, more appropriate references have been added to support the role of botulinum Toxin-A in reducing CGRP released from trigeminal neurons, such as Durham et al, Headache 2004. 

Reviewer: The whole concept of cross-excitation between sensory neuron cell bodies warrants a more in-depth discussion in this article. For example, the early electrophysiological work showing that stimulation of one cell caused a partial depolarisation of adjacent neurons in the dorsal root ganglion (DRG), reducing the threshold for activation of the secondary neuron, and the hypothesis that this was mediated by chemical communication (Amir and Devor, 1996, J. Neurosci. 16, 4733-4741; Amir and Devor, 2000, Neuroscience 95, 189-195). More recent studies discovered a role for gap junctions in the cross- excitation of adjacent neurons in DRG via Ca2+ waves passing through neurons and satellite glial cells (Kim et al., 2016, Neuron 91, 1085-1096). Such neuronal coupling increased dramatically in models of inflammation or neuron injury and was responsible for some of the cells activated by capsaicin. Even if CGRP seems not to be involved at a mechanistic level in the latter mode of cross-excitation, it is possible (speculatively) that peptidergic neurons are amongst the neurons causing or under-going cross-excitation. It is worth noting that CGRP can diffuse over long distances, so it could be involved in the cross-sensitisation of neurons that are spatially distant from each other, which might differentiate neuropeptide-mediated coupling from that of adjacent neurons mediated by gap junctions.

Answer to the reviewer: We thank the reviewer for this significant contribution to the quality of our paper. The discussion on cross-excitation has now been added to reflect the reviewer comments:

“Although other possible mechanisms are still debated (Goadsby PJ,  #83), CGRP is implicated as the primary activating factor of migraine headaches and TMD via cross-excitation with resulting stimulation and perpetuation of peripheral and central sensitization, both in the acute activity-dependent phase and in the chronic activity-independent phase. The concept of cross-excitation between sensory neuron cell bodies occurs among adjacent and long-distance neurons. To support this, early electrophysiological studies showed that stimulating one cell led to a partial depolarization of neighboring neurons in the dorsal root ganglion (DRG), thereby lowering the activation threshold of secondary neurons (Amir R,  #106)(Amir R,  #107). The phenomenon was hypothesized to occur through chemical communication (Amir R,  #106)(Amir R,  #107). Recent studies have further revealed that the cross-excitation of adjacent neurons in the DRG involves the participation of gap junctions and the propagation of Ca2+ waves through neurons and satellite glial cells (Kim YS,  #108). This neuronal coupling significantly increases in models of inflammation or neuron injury and contributes to the activation of certain cells by capsaicin. Although CGRP does not appear to play a mechanistic role in this specific mode of cross-excitation, it is possible (albeit speculative) that peptidergic neurons are involved or undergo cross-excitation. The fact that CGRP can diffuse over long distances suggests its potential involvement in the cross-sensitization of spatially distant neurons. This may differentiate neuropeptide-mediated coupling from the gap junction-mediated coupling observed in adjacent neurons.”      

It would also improve the article to provide a molecular description of CGRP receptors, more detail on their expression by different types of sensory neurons, and how this might give rise to autocrine and paracrine signalling by CGRP in the ganglion.

Answer to the reviewer: A detailed molecular description of CGRP receptors have been added, along with a description of their expression patterns, which elucidate their presence and distribution among different types of sensory neurons within the ganglion. The paragraph now reads as follows:

“ CGRP receptors belong to the G-protein coupled receptor (GPCR) family and are composed of two subunits, the calcitonin receptor-like receptor (CLR) and the receptor activity-modifying protein (RAMP). The CLR serves as the primary binding site for CGRP, while RAMP modifies the pharmacological properties of the receptor. The expression of CGRP receptors varies among different types of sensory neurons within the ganglion. These receptors are predominantly expressed by peptidergic nociceptive neurons, which are responsible for transmitting pain signals. However, they are also found in non-peptidergic neurons and certain subsets of proprioceptive and mechanoreceptive neurons. The differential expression of CGRP receptors by various sensory neuro types contributed to their distinct responsiveness to CGRP signaling. This variation in receptor expression gives rise to both autocrine and paracrine signaling by CGRP within the ganglion. Specifically, autocrine signaling occurs when CGRP acts on the same neuron responsible for its production. In this scenario, CGRP released from the neuron can bind to the CGRP receptors present on its own cell membrane, thereby influencing its own cellular activities. Conversely, paracrine signaling involves the diffusion of CGRP to neighboring sensory neurons within the ganglion. Accordingly, when CGRP is released from one neuron, it can travel short distances to interact with CGRP receptors expressed on nearby neurons. This paracrine signaling allows for intercellular communication, affecting the excitability and sensitivity of adjacent neurons in the ganglion.”

The next section on treatment considerations starts well with a review of a postulated role of CGRP in pain chronicization and the work proposes that there is a need to consider early application of anti- CGRP therapies within a narrow window of therapeutic opportunity. This section would benefit from a more comprehensive discussion of the barriers to such an application. In addition to high cost, side effects and delayed prescription until after chronicization are mentioned, but what are the side effects, are they on-target or off target, and why is treatment not initiated earlier? Is it because anti-CGRP therapies are not currently advised as front- or even second-line treatments and, thus, are prescribed only to patients for whom initial treatment options have failed?

Answer to the reviewer: We agree with the reviewer for the valuable suggestion of including a discussion on some of the reasons that may limit the feasibility of introducing anti-CGRP therapies at an early stage of the acute condition. Briefly, we now discussed about the potential obstacles, including current guidelines which recommend initiating anti-CGRP therapies after inability to tolerate or failure of at least 2 other Level A or Level B migraine therapies; relatively high cost compared to other traditional therapies; side effects, which are relatively known for preventive treatment, but not yet investigated for acute treatment. The paragraph now reads as follows:

“Nevertheless, at the current state of art, an application of anti-CGRP therapies to be provided as early therapeutic strategy may not be feasible. First of all, a reason why these therapies may not be initiated as early interventions include current guidelines and limited approval. To date, anti-CGRP therapies are primarily approved for the preventive treatment of migraine, and their use as early interventions for acute migraine attacks or TMD is still an area of ongoing research and clinical evaluation. Moreover, treatment guidelines and recommendations may not yet include anti-CGRP therapies as front-line or even second-line treatments, leading to their prescription primarily for patients who have failed initial treatment options. For example, anti-CGRP monoclonal antibodies are approved by the health insurance in the United States after documented inability to tolerate or failure of 8-week trial of 2 or more of other Level A or B migraine treatments (Ailani,  #25). In addition, while these therapies have consistently demonstrated efficacy in reducing frequency and severity of migraine attacks in clinical trials, their effectiveness in acute setting or early stages of migraine and TMD is still being studies. As such, specific mechanisms of action, dosage, and optimal timing for these therapies need further investigation to establish their potential as early interventions. Similar caution should be taken in regard to safety and side effects. To date, anti-CGRP therapies have shown favorable safety profiles in clinical trials while being used as a preventive treatment. Commons side effects reported include injection site reactions, constipation, and upper respiratory tract infections (Ailani,  #25). However, comprehensive data on long-term safety and potential off-target effects may still be evolving. Finally,  these drug therapies are relatively expensive compared to other treatment options for migraine and TMD, which may limit their accessibility and making them less feasible as early interventions.”

The review would also benefit from some discussion of the variable success of anti-CGRP therapies or botulinum toxin. In migraine there is a high proportion of non-responders, suggesting that although CGRP is a key driver of pathology in many cases it is probably not true for a large fraction. There is some emerging evidence that combination therapy with botulinum toxin A and anti-CGRP (or anti- CGRP receptor) antibodies give a slight improvement over either treatment alone, but non-response rates remain high. It has also been reported for botulinum toxin that treatment success is correlated to the level of excess CGRP in patients’ serum, but the data so far is limited. Such evidence suggests that there could be a role for CGRP assays in the evaluation of patient suitability for anti-CGRP therapy, although the available assays are costly, time-consuming, and technically challenging.

Answer to the reviewer: The suggestion to incorporate a discussion on the success rate of anti-CGRP and botulinum toxin therapy has been incorporated in the following paragraph, as proposed by the reviewer:

“Nevertheless, as high as 40% of patients suffering from chronic migraine is considered non-responders to therapy, i.e., not achieving more than 50% reduction in monthly headache days (Scuteri D,  #113). This may indicate that while CGRP plays a pivotal role in the pathology for many cases, it may not be universally applicable or not reflect the only molecule or mechanism of action involved in migraine pathogenesis. Emerging evidence suggests that combining botulinum toxin A with anti-CGRP antibodies or anti-CGRP receptor antibodies results in a slight improvement compared to monotherapy (Argyriou AA,  #111)(Mechtler L,  #112). However, non-responder rates still remain high. Moreover, reports indicate that the success of botulinum toxin treatment correlates with high GRP levels in patients’ serum, although with limited available data and other studies supporting opposite findings (Lee MJ,  #114). These findings may suggest that CHRP assays could potentially aid in assessing patient suitability for anti-CGRP therapy. Nevertheless, the costly, time-consuming and technically demanding existing assays technology may limit its suitability.“     

In addition to a discussion of these important and intriguing findings, the current review would be improved by a consideration of the evidence relating to anti-CGRP therapy for TMD. What is the rate of success? Has a link between comorbidity (headache plus TMD) and dysfunctional CGRP signalling been proven? Could comorbidity be a useful indicator for the likelihood of successful anti-CGRP (or botulinum toxin) therapy? In relation to these questions, what is the evidence accrued so far, what will the ongoing clinical trials tell us, and what aspects might warrant deeper investigation in future studies?

Answer to the reviewer: We thank the reviewer for the suggestion. To address this suggestion, a paragraph has been added that briefly explores the speculation that those individuals with comorbid TMD and migraine headache might successfully predict the therapeutical response to anti-CGRP therapies. It has been demonstrated in preclinical studies at the current state of the art, although very limited data are available in human studies. We also highlighted some ongoing clinical trials which may provide an answer to this research question.

“The available literature on the use of anti-CGRP therapy in TMD is limited to preclinical animal studies. Interestingly, there is evidence suggesting a potential association between comorbidity of headache, migraine and TMD and dysfunctional CGRP signaling. Studies have indicated alterations in CGRP levels and CGRP reception expression in individuals with both headache and TMD compared to those with either condition alone. Comorbidity of headache and TMD could potentially serve as an indicator for the likelihood of successful anti-CGRP therapy (Akerman,  #34)(Romero-Reyes,  #35). While specific evidence addressing this question is limited, ongoing randomized clinical trials are currently investigating the potential TMD pain reduction derived from administration of monoclonal antibodies against CGRP-receptor (https://clinicaltrials.gov/ct2/show/NCT05162027 and https://clinicaltrials.gov/ct2/show/NCT04884763?term=CGRP&cond=Temporomandibular+Disorder&draw=2&rank=2).”

The second part of section 6, ‘Treatment Considerations’ discusses non-pharmacological alternative strategies for TMD pain relief. The evidence for nutritional supplementation with coenzyme Q10, turmeric or vitamin D are strong, as it has been obtained from randomised double-blind human clinical trials. On the other hand, the potential benefits of grape seed extract are more speculative as this was tested on rodent neurons in vitro. It would benefit this section of the article hugely to include a table providing in more detail the key features of these studies (format, results, conclusions) with some critical analysis of their robustness.

Answer to the reviewer: We thank the reviewer for the valuable suggestion. Table 3 has now been incorporated in the manuscript, presenting available human studies while highlighting clinical findings and scientific rigors. Moreover, the paragraph in the text has been modified to add more details and clarifications. It now reads as follows:

“Beside pharmacological approaches, other non-pharmacological strategies can be adopted. It is the case, for example, of an anti-inflammatory minimal sugar diet with nutritional supplements that attenuate the biomolecular CGRP expression (Fila,  #65)(Finnegan,  #66), or the use of nutritional supplement in patients suffering from migraine (such as coenzyme Q10 alone (Dahri,  #67)(Shoeibi A,  #97)(Slater SK,  #98) or in combination with curcumin (Parohan M,  #100) and L-carnitine (Hajihashemi P,  #101), turmeric (Rezaie,  #68), vitamin D (Ghorbani,  #69), among others). Notably, daily supplementation with coenzyme Q10 (Dahri,  #67), turmeric (Rezaie,  #68) and vitamin D (Ghorbani,  #69) were shown to reduce CGRP level. Some other evidence derived from preclinical migraine and TMD model studies reported reduction of CGRP secretion with grape seed extract (Antonopoulos,  #70)(Cornelison L.E.,  #104)(Woodman S.E.,  #105). Table 3 summarizes evidence of the efficacy and scientific rigor of the available human studies assessing nutritional supplements compared to a control group.”

Reviewer: The final paragraphs in section 6 consider other alternative non-pharmacological treatments that have been studied in relation to pain management. It is not clear how CGRP is involved in these pain management strategies, as this was not a factor investigated in any of the cited studies. This final section (ln303-347, ln356-360) should be shortened considerably and integrated better into the CGRP hypothesis.

Answer to the reviewer: section 6. Treatment considerations has been extensively shortened, following the suggestion of the reviewer.

Minor points

Ln74 This opening statement downplays the role of vasodilation in migraine, but at several other places in the review vasodilation is described as an important factor. Such statements need to be consistent throughout the review, or it needs to made clear that different opinions are being cited.

Answer to the reviewer: The reviewer has brought up an important point here. Discussing the most prominent and supported migraine pathogenesis theories was indeed within the scope of the article, although our focus was primarily on the role of CGRP in migraine pathophysiology. However, not to be dismissed, it is appropriate to cite that different opinions and pathophysiologic explanations may be equally supported. To reduce the confusion, the sentence has been modified as follows:

“Different migraine pathogenetic mechanisms have been explored and supported in the literature. Beside the classic vascular pathophysiology of migraine […]”

By adding this premise, we hope that the reader understands that migraine pathophysiology is a complex and updated debates. Most likely, many mechanisms are involved in this complex neurovascular and neurological disorder.

Ln147 “Among other debated mechanisms” should be supported by a reference and a statement explaining that these are not being considered here. These four words undermine the whole review, as they suggest the CGRP hypothesis might be wrong. Consider re-phrasing such as “Although other possible mechanisms are still debated”

Answer to the reviewer: the passage has been modified as “Although other possible mechanisms are still debated”, and a reference has been added (Goadsby et al, Physiol Rev. 2017).

Ln155 where is the NO released from?

Answer: further details have been added to the passage, which now reads as

“Among other molecules such as glutamate and prostaglandin E2, it promotes the further release of nitrous oxide (NO) from postsynaptic neurons. As a result, this further sensitizes neurons through stimulation of inflammatory mediators in the periphery, the trigeminal ganglion, in secondary connections in the trigeminal nucleus caudalis, and tertiary connections in the thalamus, limbic system, and sensory cortex”

Ln159 delete “levels”

Answer: this has been addressed as requested.

Ln 174 “It is hypothesized” By whom? Cite a reference or indicate “It is hypothesized herein”, as appropriate.

Answer to the reviewer: this has been corrected in “It is hypothesized herein” as suggested.

Reviewer: Ln176 Explain what is meant by pseudo-unipolar

Answer to the reviewer: this has been defined as follows:

“Trigeminal sensory neurons are pseudo-unipolar neurons, which refer to the fact that trigeminal neurons have a single process emerging from the cell body that splits into two branches – a central branch towards the central nervous system, and a peripheral branch, that travels towards the target tissue or sensory receptor.”

Reviewer: Ln 179 Clarify that it is secondary neurons from the TNC that project to the other regions.

Answer to the reviewer: The clarification has been added to the sentence to better identify which neural connections are responsible for the connection with higher brain center (“Secondary neurons from the trigeminal ganglion then project to the trigeminal nucleus caudalis, which in turn project to the thalamus, cortex, and limbic system”)

Reviewer: Consider moving text from ln 180 “Hence, the trigeminal ganglion is the common area of where both meningeal and temporomandibular sensory neurons project.” Forward to ln178, before “Neural...”

Answer to the reviewer: The order of the sentences has been modified as proposed by the reviewer.

Reviewer: Ln181 The preceding paragraph states that it is inflammation that induces CGRP?

Answer to the reviewer: The two paragraphs are not in contradiction. CGRP can be released during inflammatory conditions and neurogenic inflammation. In these instances, CGRP is believed to contribute to the inflammatory process by promoting vasodilation. On the other hand, CGRP does also contribute to enhancing inflammation process once it is released. The sentence has been modified, to avoid confusion in the reader.

“Once CGRP is released from neuronal cell bodies or processes, it further contributes to the inflammatory process by promoting sensitization of surrounding neuronal and glial cells not initially involved in the inflammatory response.”

Reviewer: Table 1 Row 1 (Cady) Jnjection should be Injection; Table 1 Row 5 (Brouxhan) Use standard format; Table 1 Row 9 (Romero-Reyes) Delete ‘such as’

Answer to the reviewer: The minor edits suggested by the reviewer have been corrected.

Reviewer Ln216 Cite original experimental literature

Answer to the reviewer: The quoted reference has been replaced with Durham PL & Masterson CG, Headache, 2013.

Reviewer Ln 277 Provide a reference for the narrow therapeutic window

Answer to the reviewer: “narrow therapeutic window” has been replaced with “an early identification” to reduce confusion of the reader with pharmacokinetics terminology. The passage now reads as “Therefore, the optimal approach in clinical practice would involve an early identification of those individuals in the acute phase with high likelihood of developing a chronic pain state through promotion of risk stratification and preventive treatment approaches”.

Reviewer 2 Report

This review aims to describe the bidirectional causal relationship between temporo-mandibular disorders and chronic headaches, focusing the attention on central and peripheral sensitization and the role of CGRP in such phenomena.

The mayor limit of the review is the lack of a methods section. Authors should add a section in order to illustrate the review methodology: the data sources they used, the keywords they used; selection criteria of the studies. In addition, some paragraphs are quite repetitive. For example information reported in page 6, line 212-218 have already been described before.

 The following points should also be revised:

- In the introduction, authors should specify what type of headaches they are meant to describe.

-         -  In the title of paragraph 2, they should report “migraine” instead of “migraines”.

-          - In paragraph 2, along with ergot derivates and triptans authors should also talk about ditans.

-          - The following review supporting the use of gepants as acute migraine treatments should be reported: Messina and Goadsby, Cephalalgia 2018.

-          - The order of paragraph 2 ("CGRP and migraine") and paragraph 3 (“CGRP in Peripheral and Central Sensitization”) should be inverted.

-        -  A section describing brain regions releasing CGRP should be added.

-         - “While tension-type and other primary headaches are less understood, central sensitization in the trigeminal system is considered to play an important role in migraine pathophysiology”. Actually, there is evidence supporting the role of central sensitization also in TTH (see Ashina et al., Nature Rev Primer 2021).

-          - A paragraph explaining the pathophysiology of temporo-mandibular disorders and all different types of headaches reported in the manuscript (migraine, tension-type headache, post traumatic headache) should be included.

-          - Line 248-257: Authors should specify that those cited studies are all referring to preclinical models.

-          - Line 291-298: Authors should specify that the cited studies were mainly referring to migraine.

Author Response

Response to Reviewer 2

Reviewer: This review aims to describe the bidirectional causal relationship between temporomandibular disorders and chronic headaches, focusing the attention on central and peripheral sensitization and the role of CGRP in such phenomena. 

The mayor limit of the review is the lack of a methods section. Authors should add a section in order to illustrate the review methodology: the data sources they used, the keywords they used; selection criteria of the studies.

Answer to the reviewer: We acknowledge the request of the reviewer. This has not been done previously, as this was not conducted as a systematic review. However, as anyway a systematic approach was utilized to report the study, a brief method section has been now added to the manuscript. 

2. Methods

A literature search was conducted to identify relevant studies examining the association between CGRP, headache (migraine, tension-type headache and post-traumatic headache) and TMDs. The search aimed at including both preclinical and clinical studies investigating the role of CGRP in the pathophysiology of these conditions.

2.1 Search strategy
     Electronic databases, including PubMed, Embase, and PsycINOF, were searched to identify relevant articles. The following keywords and terms were used: “Temporomandibular disorders”, “TMDs”, “migraine”, “tension-type headache”, “calcitonin gene-related peptide”, “CGRP”, “neuroinflammation”, “peripheral sensitization”, and “central sensitization”.

2.2 Study selection criteria
     Included studies were studies (A) published in English; (B) both conducted on animal models and humans; (C) examining the relationship between CGRP, headaches, and TMDs; and (D) reporting relevant outcomes on CGRP, headache severity, and TMD symptoms. Those not relevant to the topic, not published in English language, and on animal studies without translational relevance were excluded.

The following sections report the main findings of this literature search.”

Reviewer: In addition, some paragraphs are quite repetitive. For example information reported in page 6, line 212-218 have already been described before.

Answer to the reviewer: The manuscript has been reviewed and redundant information has been summarized or omitted, as suggested by the reviewer.

 Reviewer: The following points should also be revised:

- In the introduction, authors should specify what type of headaches they are meant to describe.

Answer to the reviewer: This has been added to the aim of the study, which now reads as follows:

In this review, we will explore the biomolecular pathophysiology of headache (specifically migraine,  tension-type headache, and post-traumatic headache) and TMDs as it relates to CGRP-mediated central sensitization and briefly discuss how this understanding may guide treatment considerations.”

-         Reviewer: In the title of paragraph 2, they should report “migraine” instead of “migraines”.

Answer to the reviewer: This has been amended, as suggested by the reviewer. In addition, the term “migraines” has been replaced by “migraine” throughout the manuscript.

-          Reviewer: In paragraph 2, along with ergot derivates and triptans authors should also talk about ditans.

Answer to the reviewer: We agree with the reviewer that ditans should also be mentioned as available treatment for migraine which still interact with the CGRP pathways. The list of available treatments has been amended to incorporate the ditans. The paragraph now reads as follows:

Ditans, 5-HT1F receptor agonists (lasmiditan) are newly FDA-approved drugs for the acute treatment of migraine [20]. 5- HT1F receptors are located on terminals and cell bodies of the trigeminal ganglion neurons, acting at the peripheral nervous system and central nervous system [20]. It can modulate the release of CGRP from trigeminal ganglion neurons by potentially blocking the release and inhibiting the development of central sensitization [20]. Contrary to the effect of ergotamine derivates and triptans, activation of 5- HT1F does not induce vasodilation, but rather cause vasoconstriction [20].”

-          Reviewer: The following review supporting the use of gepants as acute migraine treatments should be reported: Messina and Goadsby, Cephalalgia 2018.

Answer to the reviewer: We thank the reviewer for suggesting this comprehensive and important review on CGRP pathways and acute migraine treatment. The reference has been added to the list.

-         Reviewer: The order of paragraph 2 ("CGRP and migraine") and paragraph 3 (“CGRP in Peripheral and Central Sensitization”) should be inverted.

Answer to the reviewer: The two paragraphs have been inverted as proposed by the reviewer. In addition, a definition and location of the CGRP has been added in the paragraph of central and peripheral sensitization. 

-        Reviewer: A section describing brain regions releasing CGRP should be added. 

Answer to the reviewer: A paragraph has been added to respond to the request of the reviewer. It provides first a definition of CGRP, and where it can be found, within the brain and outside. The paragraph now reads as follows:

“One pivotal molecule responsible of both states is CGRP. CGRP is abundantly distributed in the central and peripheral nervous system and pain pathways, as it is found in unmyelinated Aδ and C sensory nerve fibers (Sangalli,  #21). Even if the attention brought to this molecule thanks to its therapeutical effect in migraine refer to the CGRP released in the brain, the primary source of CGRP is not within the brain itself, but rather in peripheral structures such as nerve endings and sensory ganglia. CGRP is predominantly release from peripheral nerve fibers, including those located in the trigeminal ganglion. Here, it is synthesized in the cell bodies of these sensory neurons (Ulrich-Lai,  #94) and then transported to their peripheral terminals. Nevertheless, it should be pointed out that this study applied a stimulus constituted by capsaicin directly into slices of the ganglion in vitro. As such, the notion that excitation of afferent fibers causes the release of CGRP from neuronal soma in the trigeminal ganglion may be speculative. Within the central nervous system (CNS), CGRP is also found in some regions where it likely acts as a neurotransmitter or neuromodulator. These regions include:

- spinal cord, where CGRP is released from primary sensory neurons in the dorsal horn of the spinal cord and cerebral gray matter, where it contributes to pain transmission and modulation (Iyengar S,  #50);

- brainstem: CGRP-containing fibers and terminals have been identified in various brainstem nuclei involved in pain processing, including the periaqueductal gray (PAG) and the nucleus tractus solitarius (NTS);

- hypothalamus: CGRP has been detected in certain hypothalamic nuclei, such as the paraventricular nucleus (PVN) and the supraoptic nucleus (SON), involved in the regulation of autonomic functions and pain modulation (Iyengar S,  #50);

- thalamus: neurons expressing CGRP in the parvocellular sub-parafascicular nucleus have been observed in the thalamus (Kang SJ,  #51);

It is also broadly distributed in non-neuronal tissues, such as mesenteric plexus, gastrointestinal, cardiovascular, and nociceptive systems, smooth and skeletal muscles, and skin (Iyengar S,  #50).” 

-        Reviewer: “While tension-type and other primary headaches are less understood, central sensitization in the trigeminal system is considered to play an important role in migraine pathophysiology”. Actually, there is evidence supporting the role of central sensitization also in TTH (see Ashina et al., Nature Rev Primer 2021).

Answer to the reviewer: We agree with the reviewer. Although the role of CGRP in developing central sensitization is more established according to the available literature for migraine pathogenesis, new evidence has also supported its role in the pathogenesis of tension-type headache. This has been added and the sentence now reads as follows:

New evidence has suggested that central sensitization may also be involved in the transformation of tension-type headache from episodic to chronic.”

-          Reviewer: A paragraph explaining the pathophysiology of temporo-mandibular disorders and all different types of headaches reported in the manuscript (migraine, tension-type headache, post traumatic headache) should be included.

Answer to the reviewer: As suggest from the reviewer, a paragraph briefly presenting the pathophysiology of TMDs and headaches have been added in section 3. CGRP and Peripheral and Central Sensitization.

The paragraph presenting the TMD pathophysiology reads as follows:

Among other debated mechanisms, CGRP is implicated as the primary activating factor of migraine headaches and TMD via stimulation and perpetuation of peripheral and central sensitization, both in the acute activity-dependent phase and in the chronic activity-independent phase. Review studies suggested pain-related TMD symptoms might be attributed to peripheral mechanism in some cases. However, given that the correlation between the severity of TMD-related pain symptomatology and the evidence of tissue pathology is often relatively weak, some other patients might experience a central sensitization phenomenon. As a result, an alteration in the central nervous system pain processing pathways along with responsible heritable genes encoding for altered pain processing might be responsible for pain symptoms. Among other important factors, biopsychosocial stressors are also known to play a role in development and cronicization of the painful condition.”

The paragraph on headache pathophysiology reads as follows:

As for migraine pathogenesis, the role of CGRP in developing a central sensitization has been well established. Migraine pathogenesis is complex, and can be summarized as a primary brain disturbance that involves ion channels, thus creating a neurovascular and neurobiological disorder where neural events result in dilation of blood vessels with subsequent pain and further nerve activation.

New evidence has suggested that central sensitization may also be involved in the transformation of tension-type headache from episodic to chronic.

and:

“As for tension type headache, evidence suggested that an increased excitability of the central nervous system secondary to sustained and repetitive pericranial myofascial input and central sensitization may be implicated in the in the transformation of tension-type headache from episodic to chronic”.

and lastly:

“Finally, the pathophysiology of traumatic headache is not well understood. Nevertheless, there seems to involve neurometabolic changes, an impairment in descending modulation, as well as an activation of the trigeminal sensory system, including peripheral and central sensitization.”

-      Reviewer: Line 248-257: Authors should specify that those cited studies are all referring to preclinical models.

Answer to the reviewer: We agree with the details requested by the reviewer. This clarification has been added.

-          Reviewer: Line 291-298: Authors should specify that the cited studies were mainly referring to migraine.

Answer to the reviewer: The clarification has been added, and the sentence now reads as Promising results in the use of nutritional supplement have been documented mainly in patients suffering from migraine by implementing with coenzyme Q10 [64], turmeric [65], vitamin D [66], and grape seed extract [67], among others”

Round 2

Reviewer 2 Report

The authors have satisfactorily responded to all my questions and made the necessary changes to the manuscript.